# Exploratory Review and In Silico Insights into circRNA and RNA-Binding Protein Roles in γ-Globin to β-Globin Switching

**DOI:** 10.3390/cells14040312

**Published:** 2025-02-19

**Authors:** Alawi Habara

**Affiliations:** Department of Biochemistry, College of Medicine, Imam Abdulrahman Bin Faisal University, P.O. Box 1982, Dammam 31441, Saudi Arabia; ahhabara@iau.edu.sa; Tel.: +966-50483-0614

**Keywords:** β-globin gene cluster regulation, hemoglobin regulation, circular RNA, RNA binding proteins, sickle cell disease, β-thalassemia,

## Abstract

β-globin gene cluster regulation involves complex mechanisms to ensure proper expression and function in RBCs. During development, switching occurs as γ-globin is replaced by β-globin. Key regulators, like BCL11A and ZBTB7A, repress γ-globin expression to facilitate this transition with other factors, like KLF1, LSD1, and PGC-1α; these regulators ensure an orchestrated transition from γ- to β-globin during development. While these mechanisms have been extensively studied, circRNAs have recently emerged as key contributors to gene regulation, but their role in β-globin gene cluster regulation remains largely unexplored. Although discovered in the 1970s, circRNAs have only recently been recognized for their functional roles, particularly in interactions with RNA-binding proteins. Understanding how circRNAs contribute to switching from γ- to β-globin could lead to new therapeutic strategies for hemoglobinopathies, such as sickle cell disease and β-thalassemia. This review uses the circAtlas 3.0 database to explore circRNA expressions in genes related to switching from γ- to β-globin expression, focusing on blood, bone marrow, liver, and spleen. It emphasizes the exploration of the potential interactions between circRNAs and RNA-binding proteins involved in β-globin gene cluster regulatory mechanisms, further enhancing our understanding of β-globin gene cluster expression.

## 1. Introduction

The regulation of hemoglobin (Hb) expression follows a well-orchestrated developmental process, transitioning from embryonic to fetal, and ultimately to adult, forms [1]. During early embryogenesis, Hb synthesis initiates in the yolk sac, where primitive erythroid cells produce embryonic hemoglobins, including Hb Gower 1, Hb Gower 2, and Hb Portland [2,3,4,5,6,7]. As development progresses into the latter part of the first trimester, erythropoiesis shifts to the fetal liver, coinciding with the suppression of ζ- and ε-globin genes and a concurrent rise in α- and γ-globin expression, leading to the synthesis of fetal hemoglobin (HbF). HbF comprises γ-globin chains encoded by *HBG1* and *HBG2*, which incorporate either alanine (Aγ) or glycine (Gγ) at position 136, respectively [7]. At birth, Gγ is predominant, but postnatally, it is gradually replaced by Aγ [2,3]. By the third trimester, the bone marrow becomes the principal site of erythropoiesis, initiating the silencing of γ-globin and the upregulation of δ- and β-globin genes, facilitating the transition from HbF to adult hemoglobins, HbA1 and HbA2 [7]. In healthy adults, HbA1, consisting of two α- and two β-globin subunits, constitutes approximately 97% of total hemoglobin, while HbA2 and residual HbF contribute 2% and 1%, respectively [7,8].

The regulatory mechanisms governing γ-to-β-globin switching revolve around transcription factors that modulate the β-globin gene cluster on chromosome 11 through direct and indirect interactions [1]. Among these regulators, BCL11A serves as a critical silencer of γ-globin expression, and its role in γ- to β-globin switching has been extensively studied [1,9,10,11,12,13,14]. Advances in understanding BCL11A molecular pathways have facilitated the development of therapeutic interventions targeting HbF induction. The gene-editing therapy exa-cel, a nonviral approach utilizing CRISPR-Cas9, disrupts the erythroid-specific enhancer of BCL11A in autologous CD34+ hematopoietic stem and progenitor cells (HSPCs), thereby reactivating γ-globin expression and increasing HbF levels. This groundbreaking therapy is FDA-approved to treat both sickle cell disease and transfusion-dependent β-thalassemia, marking a significant milestone in hemoglobinopathy management [15,16,17].

The regulation of γ- to β-globin switching involves transcription factors like KLF1 and ZBTB7A, along with co-transcription factors such as LSD1 and PGC-1α, which remain active areas of research [18,19,20,21,22]. However, to broaden the current understanding of this switch, other genetic factors should be explored. In this context, circular RNAs (circRNAs) represent a relatively new area of research. Although they were discovered in the 1970s, their significance as a functional RNA molecule has only been appreciated in the last decade [23,24,25,26,27]. In particular, circRNAs are non-coding RNAs that have closed loops with no poly-adenylated tail [28]. Due to their loop structure, circRNAs are highly stable in both intracellular and extracellular environments, highlighting their potential importance as biological markers for health and disease [26,27,28]. Over the past decades, multiple studies have shown that circRNA plays an important role in gene regulation [29,30,31,32,33]. However, currently, there are limited studies linking circRNAs to β-globin gene cluster regulation, meaning further research is warranted in this area. One study reports that hsa_circRNA_100466 [34], identified as has-LBR-0002 in circAtlas and circLBR(7,8).1 as its uniform ID [35], is downregulated in β-thalassemia carriers with high HbF levels [34]. It forms a ceRNA network with miR-19b-3p to repress SOX6, a γ-globin suppressor, promoting HbF induction and highlighting its therapeutic potential in β-thalassemia [34]. Another study reported circ-0008102 [36], identified as has-LCOR-0002 in circAtlas and circLCOR(5,6,7).1 as its uniform ID [35], is significantly upregulated in pediatric β-thalassemia patients without blood transfusion [36]. It is associated with increased γ-globin expression and functions as a miRNA sponge, regulating a network of miRNAs and their target genes [36]. These characteristics suggest has-LCOR-0002 as a potential biomarker and therapeutic target in β-thalassemia [36].

Importantly, circRNA databases offer an excellent starting point for initiating circRNA research. There are multiple free-access circRNA databases, but circAtlas 3.0 is the most promising, as it contains the largest number of circRNAs identified from high-throughput RNA sequencing using both Illumina and nanopore sequencing datasets [35]. Additionally, circAtlas 3.0 reports circRNAs using uniform IDs [37] in addition to circAtlas IDs, and it also offers a conversion tool to search for circRNA nomenclature across other databases [35]. The uniform IDs are particularly useful, as they indicate the gene name and which exons the circRNA contains [35,37]. This database facilitates the search, visualization, and downloading of circRNA expression profiles, aiding research into circRNA biology across tissues, development stages, and diseases.

This review aims to explore circRNAs expressed from the β-globin gene cluster and genes encoding key regulators of γ-globin expression in tissues such as bone marrow, spleen, liver, and blood, which is crucial for several reasons. First, it enhances our understanding of the complex mechanisms involved in β-globin gene cluster regulation, specifically with the γ- to β-globin switching, potentially uncovering new regulatory pathways and interactions. Second, it identifies potential specific circRNAs and their interactions with RNA-binding proteins (RBPs) that may reveal novel targets for therapeutic interventions in hemoglobinopathies like sickle cell disease (SCD) and β-thalassemia. Furthermore, circRNAs, due to their stability and tissue-specific expression, could serve as reliable biomarkers for diagnosing and monitoring hemoglobinopathies. This review enhances our understanding of the role of circRNA in hemoglobinopathies and its potential implications for gene therapy strategies aimed at reactivating HbF production.

## 2. Key Transcription Factors Regulating γ- to β-Globin Switching

γ- to β-globin switching is a tightly regulated developmental transition that governs the shift from HbF to HbA expression. This switch ensures that different Hb subtypes are produced at distinct stages of development, optimizing oxygen transport [2,3,4,5,6,7]. Multiple transcriptional regulators and chromatin-modifying complexes drive this transition. BCL11A and ZBTB7A are key transcriptional repressors that recruit the nucleosome remodeling and deacetylase (NuRD) complex to γ-globin promoters, leading to chromatin remodeling and gene silencing [38,39]. Additionally, methyl-CpG-binding domain (MBD) proteins exhibit multiple variants, among which MBD2 and MBD3 are specifically part of the NuRD complexes involved in γ- to β-globin switching. MBD2 is part of the BCL11A-MBD2-NuRD complex, which binds to methylated CpG islands at γ-globin promoters, facilitating chromatin remodeling and transcriptional silencing, thereby repressing HbF expression. In contrast, MBD3 is part of the ZBTB7A-MBD3-NuRD complex, which contributes to β-globin activation by promoting chromatin accessibility, ensuring proper adult hemoglobin expression. A summary of key transcription factors regulating γ- to β-globin switching is provided in Figure 1.

## 3. CircRNA Formation and Function

CircRNAs are single-stranded RNAs derived from exonic regions, intronic regions, or a combination of both, and they can also originate from intergenic genomic regions [23,24,25]. The formation of circRNAs during pre-mRNA splicing is explained by both the lariat intermediate and direct back-splicing models [23,25,26,43]. In the lariat intermediate model, linear RNA is initially produced through canonical splicing, leaving behind a lariat structure that contains introns [23,24,25]. In some cases, this lariat may also include skipped exons, along with the introns. This lariat then undergoes further processing to form circRNA [23,24,25]. In contrast, the direct back-splicing model bypasses this intermediate step, producing circRNA directly [23,25,26,43]. Additionally, RNA-binding proteins facilitate circRNA synthesis by binding to intronic complementary sequences (ICS), forming a loop that promotes circRNA generation through splicing [23,25,26,28,43,44].

A single gene can generate multiple circRNA isoforms [23,24], and circRNAs can be categorized into three classes based on their composition: exonic circRNAs (ecircRNAs), intronic circRNAs (ciRNAs), and exon-intronic circRNAs (EIciRNAs) [23,45]. EcircRNAs exclusively comprise exon sequences and are primarily found in the cytoplasm, while ciRNAs are composed solely of intron sequences and are predominantly located in the nucleus. Finally, EIciRNAs, which include both exon and intron sequences, can be present in both the cytoplasm and the nucleus [23,25,45,46,47,48]. CircRNAs can function as miRNA sponges, as one or more miRNAs can bind to the circRNA containing miRNA response elements (MRE); this binding then causes the indirect regulation of gene expression through the fine-tuning of miRNA regulation. Certain circRNAs can be translated into proteins, especially if they contain internal ribosome entry sites (IRES) or m6A-induced ribosome engagement sites (MIRES), which can initiate translation in a eukaryote and are independent of the 5′cap structure and 3′ poly A tail [46,48,49,50]. CircRNAs can function as protein sponges, binding to specific proteins and preventing them from interacting with their targets [25,44]. Additionally, circRNAs can act as protein scaffolds, forming structural frameworks by bringing multiple proteins together, thereby facilitating their interaction and function [25,51]. Indeed, circRNAs can bind to RNA-binding proteins (RBPs), thus influencing gene expression, RNA stability, and protein functions [28]. In terms of their action as protein sponges, circRNAs sequester RBPs, modulating their activity and impacting various cellular processes [28].

Understanding the interactions between circRNAs and RBPs is crucial for developing insights into cell functions and disease mechanisms in molecular medicine [28]. One notable example of an interaction between circRNA and RBP is the binding of circRNAs with argonaute (AGO). In humans, the AGO protein family comprises four isoforms: AGO1, AGO2, AGO3, and AGO4 [52]. These proteins are crucial in RNA silencing processes, particularly as integral components of the microRNA-induced silencing complex (miRISC). AGO proteins bind to different classes of non-coding RNAs, including microRNAs (miRNAs), thus aiding in mRNA cleavage and translation inhibition [28,53]. Notably, AGO2 is the most highly expressed in humans and is the only isoform with endonuclease activity, making it essential for miRNA function and gene regulation through its action in silencing target mRNAs within the miRISC [52,53]. Studies have shown that circRNA reduces the ability of miRNA to bind to target mRNAs that serve as sponge molecules for the AGO protein [28]. In particular, AGO2 has been investigated for its role in γ-globin regulation, particularly in the reticulocytes, which synthesize approximately 20% of their hemoglobin after enucleation [54,55], and it has been found that AGO2 miRISC binds to γ-globin mRNA significantly more than β-globin mRNA in adult blood (AB) reticulocytes compared to core blood (CB) reticulocytes [54]. A potential method to increase HbF levels inside AB reticulocytes involves inhibiting AGO2 miRISC formation. This inhibition prevents the miRISC from binding to γ-globin mRNA, thereby increasing the levels of HbF.

Another key RBP is up-frameshift protein 1 (UPF1), which interacts with circRNAs to regulate mRNA stability, splicing, and gene expression [56]. UPF1 is essential for nonsense-mediated mRNA decay (NMD), as it targets mRNAs with premature termination codons (PTCs) and other abnormal mRNAs for degradation [56]. UPF1 also influences the splicing of pre-mRNAs, contributing to alternative splicing regulation [56]. CircRNAs can bind to UPF1, sequestering it, affecting its activities, and, thus, indirectly stabilizing specific mRNAs [57,58,59]. Conversely, some circRNAs are targets of NMD, which ensures circRNA quality control [58,59]. These interactions between RBP and circRNA add a layer of gene regulation, thus affecting cellular functions and potentially contributing to health and disease. Consequently, understanding RBP-circRNA interactions provides insights into gene regulation mechanisms and potential therapeutic avenues for conditions involving faulty mRNA regulation.

## 4. CircRNA Expressed from the β-Globin Gene Cluster

The β-globin gene cluster is located in chromosome 11p15, and it includes five genes involved in Hb production, which are arranged in the same order with which they are expressed during development: *HBE*, *HBG1*, *HBG2*, *HBD*, and *HBB* [60,61]. The region also contains one pseudogene, *HBBP1*, and LCR, which consists of five DNase I hypersensitive sites (HS) [60,61]. According to circAtlas 3.0, multiple circRNAs are expressed from the *HBG1*, *HBG2*, *HBD*, and *HBB* genes. However, there are no data regarding circRNA expression from *HBE*, *HBBP1*, and the LCR.

## 5. CircRNAs Expressed from the *HBB* Gene

The *HBB* gene is approximately 1606 base pairs in length, with three exons and two introns, and this gene codes for the β-globin protein, a component of the adult hemoglobin (HbA) found in red blood cells [62]. The *HBB* gene also expresses 44 known circRNAs [35], with 41 of them being expressed in the blood, bone marrow, spleen, and liver [35], Figure 2. The majority of *HBB* circRNAs have binding sites for AGO2 and UPF1 RNA-binding proteins [35], Figure 3A,B. AGO2 miRISC binds more to γ-globin mRNA than β-globin mRNA in AB reticulocytes compared to CB reticulocytes [54]. Multiple binding sites have been identified for AGO2 on multiple circRNAs expressed from the *HBB* gene, indicating that these circRNAs have a role in β-globin gene cluster regulation. Indeed, circRNA can significantly influence gene expression when it possesses multiple binding sites for AGO2.

One key process by which circRNA influences gene expression through AGO2 involves the circRNA sequestering AGO2 and thereby preventing it from forming the miRISC. This inhibition prevents miRNAs from binding to their target mRNAs, thereby diminishing miRNA-induced gene silencing. Additionally, AGO2 endonuclease activity is involved in the degradation of circRNAs [59]. When a circRNA sequesters AGO2, it limits the AGO2’s ability to bind and degrade other circRNAs, thereby protecting them from degradation and allowing these circRNAs to perform their regulatory functions more effectively. Consequently, circRNAs with multiple AGO2 binding sites can modulate gene expression by both inhibiting miRNA silencing and enhancing the stability and function of other circRNAs.

UPF1, a key factor in NMD, has numerous binding sites in circRNAs originating from the *HBB* gene [56,63]. A prime example of a disorder that first highlighted the medical significance of NMD is β-thalassemia, which shows a hallmark reduced level of mRNAs harboring nonsense mutations [63]. The reduced mRNAs level highlights NMD’s role in limiting the synthesis of potentially harmful C-terminally truncated polypeptides that can potentially induce a dominant negative effect [63]. The β-thalassemia mutation β^0^39, which leads to PTC, triggers NMD, leading to the decay of β^0^39 mRNA [64,65]. The use of ribosomal read-through molecules, such as G418, induces the translation of β^0^39 mRNA in erythroid cells obtained from homozygous β^0^39 thalassemia patients, leading to the production of HbA [64,65]. Supplementing NMD inhibition with a read-through inducer synergistically enhances HbA production in these patients [64,65]. The presence of UPF1 binding sites in circRNA expressed from the *HBB* gene is of interest, as these circRNAs may play a role in β-thalassemia, particularly those circRNAs that are due to nonsense mutations. Additional studies are required to comprehensively uncover the mechanisms through which these circRNAs interact with UPF1, their impact on NMD, and the potential therapeutic implications for β-thalassemia treatment. Specifically, studies should focus on identifying which circRNAs are most effective in modulating UPF1 activity, the conditions under which these interactions occur, and how these interactions influence the overall stability and translation of mRNAs in erythroid cells. Additionally, the potential of combining circRNA-based therapies with existing NMD inhibitors and read-through inducers should be explored.

## 6. CircRNAs Expressed from the *HBD* and *HBG* Genes

According to the circAtlas 3.0 database, only one circRNA is expressed from the *HBD* gene; hsa-HBD_0001. This circRNA is uniquely expressed in the bone marrow and has several binding sites with UPF1 and AGO2 [35].

*HBG1* and *HBG2* genes are almost identical, consisting of three exons and two introns, and these genes code for the Aγ-globin and Gγ-globin proteins, respectively [7]. Both of these genes have nearly identical sequences, which makes it challenging to analyze and align the RNA transcriptome to either one. Nevertheless, according to the available data from circAtlas 3.0, the majority of circRNAs are expressed from the *HBG2* gene [35]. In particular, the *HBG2* gene expresses 46 known circRNAs across various tissues, with 23 of these circRNAs expressed in the blood, bone marrow, spleen, and liver (Figure 4) [35]. Additionally, circRNA expressed from the *HBG2* gene seems to have a broader interaction with RBP than circRNA expressed from the *HBB* gene. For simplification, this section focuses on circRNAs that have binding sites for RBPs implicated in γ- to β-globin switching, namely AGO2, insulin-like growth factor 2 mRNA-binding protein 1 (IGF2BP1), IGF2BP2, and LIN28B (Figure 5). AGO2 was discussed earlier in this work; this section will now focus on the potential roles of the other RBPs and their interactions with circRNAs in the γ- to β-globin switch.

The IGF2BP family of RBPs is essential for post-transcriptional gene regulation and consists of three main paralogs: IGF2BP1, IGF2BP2, and IGF2BP3 [66,67]. IGF2BP1 and IGF2BP3 are expressed in cord blood and fetal liver erythroblasts, but they are expressed at a low level in adult erythroblasts [66,67]. In contrast, IGF2BP2 is expressed at the same level in both cord blood and adult erythroblasts [66,67]. The overexpression of IGF2BP1 in primary human CD34^+^-derived erythroid cell cultures increases the level of HbF compared to control [66,67]. Since circRNAs expressed from *HBG2* have a binding site for IGF2BP, this suggests that they might play a role in the regulation of γ- to β-globin switching. It is possible that these circRNAs can act as a sponge for IGF2BP, thus preventing IGF2BP RBPs from binding to their targets. In contrast, these circRNAs may also act as scaffolds to form functional complexes and facilitate the assembly of multiple protein complexes. Overall, further research is needed to determine the extent to which these circRNAs fine-tune the regulation for γ- to β-globin switching.

Another RBP, LIN28B, is expressed in a developmental and stage-specific manner throughout erythropoiesis [14] and inhibits the let-7miRNA family [1,68,69]. Independently of LIN28B, the tough decoy (TuD) inhibition of let-7 miRNA upregulates HMGA2, leading to the downregulation of BCL11A and ultimately inducing γ-globin expression [69]. The overexpression of LIN28B in adult erythroblast cultures reduces the expression of let-7 miRNA and significantly increases HbF expression compared to control [68]. Additionally, LIN28B binds directly to *BCL11A* mRNA, inhibiting the transcription of this gene and, thus, increasing HbF expression [14]. LIN28B binding sites in circRNAs derived from the *HBG2* gene suggest a potential role in Hb regulation. However, further validation through functional assays to determine circRNAs influence on HbF levels and in vivo studies to assess physiological relevance is required.

It should be noted that the SNP rs7482144, also referred to as Xmn1 and known to have a strong association with HbF level, resides within the promoter region of the *HBG2* gene [1,70], making it unlikely to be included in the circRNA sequences expressed by *HBG2*. Nonetheless, it is not yet clear whether this SNP could impact the expression of circRNA from the *HBG2* gene. Understanding the potential influence of the Xmn1 SNP on the expression of circRNA from HBG2 is crucial because it could reveal additional layers of hemoglobin regulation specific to Xmn1 that could aid in further understanding its association with the HbF level in patients.

## 7. CircRNAs Expressed from the *BCL11A* Gene

The *BCL11A* gene resides on chromosome position 2p16.1, comprises four exons and three introns, and encodes a zinc-finger protein that primarily functions as a transcription repressor factor [71,72]. BCL11A is the master regulator for γ- to β-globin switching (HbF to HbA), a process that occurs during late fetal development and continues after birth [1,9,10,11,12,13,14]. Indeed, deactivating *BCL11A* increases the level of HbF in CD34^+^ erythroid cell cultures compared to control, and this effect is the basis of the recent gene therapy, exa-cel, for treating hemoglobinopathies such as SCD and β-thalassemia. CRISPR/Cas9 is used to disturb the erythroid-specific enhancer for the *BCL11A* gene ex vivo in hematopoietic stem cells, which ultimately leads to an increase in HbF and supports the treatment of SCD and β-thalassemia.

Approximately 49 circRNAs are expressed from the *BCL11A* gene, but only 22 are expressed in the blood, bone marrow, liver, and spleen (Figure 6). These circRNAs have multiple binding sites with various RBPs; however, the focus in this work is on the RBP that has a known and established role in γ-globin expression. Specifically, AGO, IG2BP2, and polypyrimidine tract-binding protein 1 (PTBP1) have binding sites for circRNAs expressed from the *BCL11A* gene (Figure 7). AGO2 and IG2BP2 were discussed earlier in this work.

PTBP1 belongs to a subfamily of heterogeneous nuclear ribonucleoproteins and features RNA-binding domains that regulate mRNA metabolism, including stability, alternative splicing, IRES-driven translation initiation, and 3′ end processing [73]. PTBP1 contributes to heme metabolism by regulating the expression of aminolevulinate synthase 2 (ALAS2), an essential enzyme in heme biosynthesis in erythroid cells [73]. Additionally, a 60% reduction in the PTBP1 protein in human CD34+ erythroid cell cultures raises HbF production fourfold compared to control [74]. Since PTBP1 is involved in the IRES-mediated initiation of the translation of circRNAs into proteins, this suggests that circRNAs derived from the *BCL11A* gene may also have the potential to be translated into proteins, and further experimental research is warranted to explore this possibility.

HbF levels are strongly associated with the SNPs rs766432, rs11886868, and rs4671393, which are situated within intron 2 of the BCL11A gene [70] and are likely to be within the sequence of some of the circRNAs expressed from *BCL11A*, namely hsa-BCL11A_0011 and hsa-BCL11A_0038, according to circAtlas 3.0. Indeed, these are intronic circRNAs, and their sequences include the position of these three SNPs. However, whether these SNPs can affect the function of these circRNAs requires further investigation.

## 8. CircRNAs Expressed from the *LSD1* Gene

Lysine-specific demethylase 1 (LSD1), also known as KDM1A, is an essential epigenetic regulator that modifies histones and non-histone proteins [75,76]. This enzyme, encoded by the LSD1 gene on chromosome 1p36.12, comprises 19 exons and 18 introns and plays a crucial role in various biological processes, including gene expression regulation, cell differentiation, and tumor progression [75,76,77,78]. In the context of β-globin gene cluster regulation, LSD1 has been shown to repress the expression of γ-globin [1,18]. The inhibition of LSD1 leads to increased HbF levels, which is beneficial in conditions such as sickle cell disease and β-thalassemia, as elevated HbF can ameliorate the symptoms by reducing the polymerization of sickle hemoglobin [18,19]. Moreover, studies have demonstrated that LSD1 inhibitors, such as RN-1, can effectively induce HbF synthesis and improve disease pathology in models of these hemoglobinopathies [19]. These findings highlight the potential therapeutic value of targeting LSD1 in the treatment of hemoglobin disorders.

Over 100 circRNAs are expressed from the *LSD1* gene, with 42 of them being expressed in the blood, bone marrow, liver, and spleen (Figure 8). These circRNAs have multiple binding sites with RBPs that are involved in γ- to β-globin switch regulation, including AGO2, PTBP1 (Figure 9A,B, respectively), IGF2BP1, IGF2BP3, PTBP1, and LIN28B (Figure 10), which have been discussed earlier in this work. Additionally, some circRNAs from the *LSD1* gene have binding sites for pumilio 1 (PUM1) and methionine aminopeptidase 2 (METAP2), (Figure 10E,D, respectively).

Firstly, PUM1 is an RNA-binding protein that has a direct role in the post-transcriptional regulation of γ- to β-globin switching without affecting the progression of erythropoiesis [79]. PUM1 has consensus binding sites in the 3′UTR of the *HBG1* gene, which are not present in the *HBG2* or *HBB* genes [79]. Knocking down *PUM1* increases γ-globin transcription and protein levels in primary CD34+ erythroid cells [79]. Multiple circRNAs expressed from the *LSD1* gene have binding sites for PUM1 (Figure 10E), and, thus, it is feasible that these circRNAs expressed from *LSD1* might have a role in hemoglobin regulation through PUM1.

Secondly, METAP2 has a primary role in protein maturation by cleaving the initiator methionine (iMet) from proteins [80]. In Hb synthesis, METAP2 cleaves the iMet from α- and β-globin, including β^s^-globin [80]. Inhibiting iMet removal from β^s^-globin delays HbS polymerization and increases oxygen affinity in red blood cells [80]. While METAP2 is not directly involved in RNA processing, it is essential for posttranslational modification. CircRNAs expressed from the *LSD1* gene may act as protein sponges, inhibiting METAP2 and influencing HbF function and stability (Figure 10D). Further investigation into these interactions may reveal new therapeutic targets for hemoglobinopathies like sickle cell disease.

## 9. CircRNA Expressed from the *PPARGC1A* Gene

Peroxisome proliferator-activated receptor-γ (PPARγ) co-activator -1α (PPARGC1A or PGC-1α) is a co-transcription factor encoded by the *PGC-1*α gene located in chromosome 4p15, which contains 13 exons and 12 introns [81]. Additionally, a novel exon has been described that is located 13.7 kb upstream of the canonical transcription start site [82]. PGC-1α is a key regulator of cellular energy metabolism, influencing various physiological processes. It contributes to thermogenesis in brown fat, supports ketogenesis and gluconeogenesis in the liver, and mediates adaptive responses in slow-twitch muscle fibers to meet energy demands [83,84,85,86,87]. Moreover, PGC-1α is involved in γ-globin expression; indeed, the overexpression of PGC-1α in CD34^+^ primary erythroid cells increases the expression of HbF [20]. ZLN005, a novel small molecule that increases PGC-1α levels in CD34+ primary erythroid cells, also increases the level of HbF in culture [20].

Approximately 63 circRNAs are known to be expressed from the *PGC-1*α gene, with 17 detected in the liver, and only one among them also found in the blood (Figure 11). Multiple RBPs have binding sites with circRNAs expressed from the *PGC-1*α gene (Figure 12), including AGO2, UPF1, and IGF2BP1, which have been discussed earlier in this work. Additionally, these circRNAs have binding sites for helicase-like transcription factor (HLTF), which increases β-globin transcription when overexpressed in the K562 cell line [88]. HLTF is not an RBP, but it is of interest that circRNA has binding sites with this transcription factor. Only hsa-PPARGC1A_0001, which is expressed in the blood and liver, has binding sites for HLTF in addition to IGF2BP1; whether this has functional significance requires experimental validation. Future studies should investigate whether HLTF binding to circRNAs affects β-globin transcription in erythroid cells. CRISPR-Cas9 could be used to edit circRNA sequences and disrupt HLTF binding sites, assessing β-globin expression changes. Overexpression or knockdown experiments in K562 cells could further clarify circRNA-HLTF interactions, though validation in CD34+ primary erythroid cells is essential. These studies would provide deeper insights into circRNA-mediated gene regulation and their therapeutic potential in hemoglobinopathies.

## 10. Limitations and Challenges Facing circRNA Research

CircRNA research has several limitations that hinder its therapeutic and diagnostic applications. One major limitation is the difficulty of selectively manipulating circRNAs without affecting their linear counterparts, as both often originate from the same transcript, making knockdown or overexpression experiments prone to unintended effects [89]. Moreover, the expression of many circRNAs is low in specific tissues, requiring large input samples and highly sensitive detection techniques for accurate quantification [90]. A critical limitation of circRNA research is the lack of standardized nomenclature and databases, resulting in inconsistencies in circRNA annotation across different studies and complicating cross-study comparisons [90]. However, this challenge can be addressed through the adoption of standardized systems, as seen in databases such as circAtlas, which implement uniform naming conventions to improve data reliability and comparability [35,37]. Functional studies are further limited by the challenge of distinguishing truly functional circRNAs from those that may simply be byproducts of splicing, as many have reported functions are based on in vitro experiments that may not completely reflect physiological relevance [91,92]. While circRNAs are known for their stability, the mechanisms governing their degradation remain poorly understood, as they require specific endonucleases rather than conventional exonucleolytic decay pathways [89]. In the field of circRNA-based therapeutics, key barriers include low translation efficiency, difficulties in efficient delivery, and potential off-target effects when using artificial circRNA constructs [93,94,95]. Moreover, the lack of efficient tools for circRNA manipulation poses a significant challenge, as the current RNA interference strategies are ineffective due to the absence of free 5′ and 3′ ends, necessitating the development of alternative knockdown techniques.

Yang et al. [34] and Chen et al. [36] emphasize the limited understanding of circRNA functions in β-thalassemia, highlighting their unclear roles in HbF regulation and disease progression [34,36]. Yang et al. identify hsa-circRNA-100466 (has-LBR-0002 in circAtlas) as a potential HbF regulator; however, they note the need for functional validation and clinical trials to confirm its role [34]. Similarly, Chen et al. report circ-0008102 (has-LCOR-0002 in circAtlas) upregulation in pediatric β-thalassemia patients; however, they highlight uncertainties regarding its miRNA interactions, regulatory mechanisms, and broader impact on disease progression [36]. Both studies stress the importance of mechanistic experiments and large-scale research to fully characterize circRNA function in β-thalassemia.

## 11. Potential Application of circRNA in Clinical Trials and Diagnostics

Despite the growing interest in circRNAs, studies investigating their roles in hemoglobinopathies and blood diseases remain limited, particularly for diagnostic and therapeutic applications. However, emerging research suggests that circRNAs could serve as biomarkers and regulatory molecules in these conditions. CircRNAs’ high stability, tissue-specific expression, and resistance to exonuclease degradation make them promising candidates for disease detection, prognoses, and therapeutic interventions [90]. They can be detected in bodily fluids, such as blood and saliva, allowing for non-invasive sample collection that can be applied to hematological disorders [89]. In hemoglobinopathies, Yang et al. [34] and Chen et al. [36] identified differentially expressed circRNAs in β-thalassemia, highlighting their roles in γ-globin regulation and their potential as non-invasive biomarkers. Expanding on these findings, Gao et al. [96] conducted a comprehensive meta-analysis on circRNAs in hematological malignancies, demonstrating their high stability and specificity as reliable biomarkers. Their findings revealed the potential of circRNAs as effective diagnostic and prognostic biomarkers in clinical settings, with an area under the curve (AUC) of 0.86 and a sensitivity and specificity of 79% in diagnostic applications [96].

Beyond diagnostics, circRNAs are being explored as therapeutic agents in clinical trials, particularly in circRNA-based RNA therapeutics [51,93]. Engineered circRNAs have been shown to serve as stable mRNA templates for protein production, offering an alternative to traditional RNA-based therapies [51,96]. A key example is the circRNA vaccine against SARS-CoV-2, which has demonstrated potent immune responses and robust protection in preclinical models, outperforming conventional mRNA vaccines in terms of antigen stability and durability [96,97]. This suggests that circRNA-based therapies can be applied to blood disorders, particularly for gene regulation and HbF induction in hemoglobinopathies.

While circRNAs present promising applications in blood diseases, challenges remain in optimizing their delivery, achieving translation efficiency, and ensuring targeted function in clinical settings. Continued advancements in bioinformatics, sequencing technologies, and RNA engineering will be crucial to maximize the clinical application of circRNAs in disease monitoring, prognoses, and targeted therapies for hemoglobinopathies.

## 12. Conclusions

The transition from γ- to β-globin is a complex process involving numerous transcription factors, co-transcription factors, and miRNAs. However, the role of circRNAs in γ- to β-globin switching has been relatively underexplored. In this review, the potential involvement of circRNAs in conjunction with RNA-binding proteins (RBPs) in γ- to β-globin switching was examined. CircAtlas 3.0 was used as a source of data regarding circRNA expression from genes known to be essential in γ- to β-globin switching, revealing potential interactions between these circRNAs and RBPs. However, further experimental research is necessary to confirm these connections. Such research could significantly benefit treatment development by uncovering novel regulatory mechanisms and therapeutic targets for hemoglobinopathies, potentially leading to more effective interventions for conditions like SCD and β-thalassemia.

## Figures and Tables

**Figure 1 cells-14-00312-f001:**
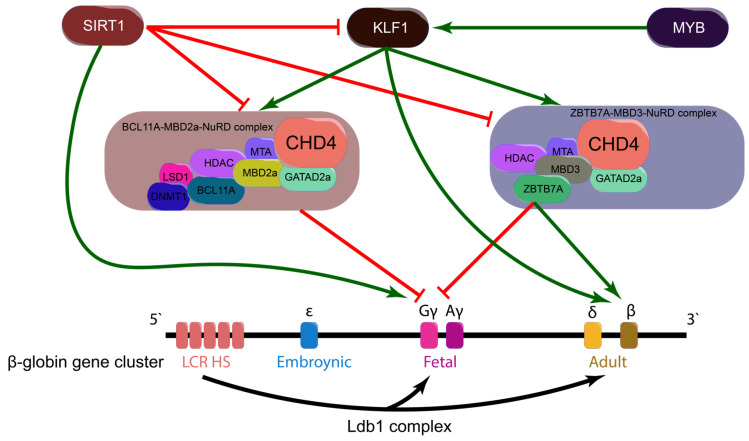
Key transcription factors affecting γ- to β-globin switching. There are two major NuRD complexes involved in γ- to β-globin switching. BCL11A-MBD2a-NuRD plays a critical role in γ-globin silencing by binding to its promoter and preventing transcriptional activation. This repression is reinforced through interactions with PRMT5, leading to a closed chromatin conformation. Additionally, LSD1 and DNMT1, which are associated with BCL11A, reinforce the γ-globin silencing by methylating the promoter region [40]. In contrast, MBD3-NuRD, which interacts with ZBTB7A, is essential for maintaining high β-globin expression by facilitating the locus control region (LCR) and *HBB* interaction. ZBTB7A has a binding motif at the γ-globin promotor region, which may indicate it also has some role in silencing the γ-globin expression. The coordinated function of these two complexes ensures the repression of HbF and activation of HbA, a process crucial for normal Hb switching. Looping between the LCR and gene promoters is facilitated by the Ldb1 complex, while MYB indirectly regulates γ-globin expression by activating KLF1 [1]. KLF1 indirectly silences γ-globin expression by stimulating BCL11A and ZBTB7A [1,13,41] and directly activates β-globin expression [12]. γ-globin expression is also stimulated directly and indirectly by silent mating type information regulation 2 homolog 1 (SIRT1) [42]. (Not drawn to scale, green arrows indicate activation; red lines indicate silencing).

**Figure 2 cells-14-00312-f002:**
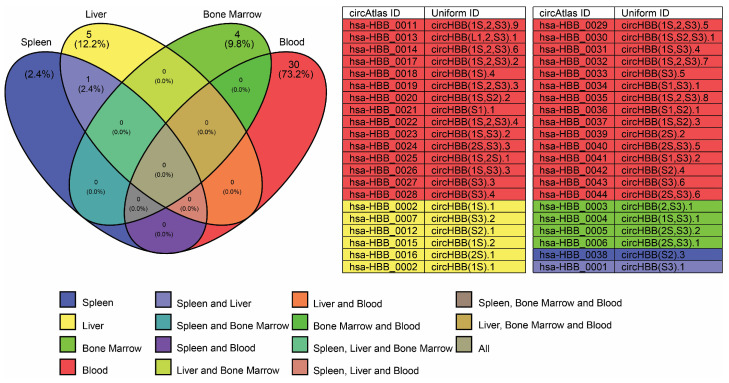
Venn diagram illustrating circRNA expression from the *HBB* gene in blood, bone marrow, liver, and spleen. A total of 41 circRNAs are expressed across these tissues, with 30 (73.2%) in blood, 4 (9.8%) in bone marrow, 5 (12.2%) in liver, 1 (2.4%) in spleen, and 1 (2.4%) shared between liver and spleen. The corresponding circRNA IDs and uniform IDs are listed in the accompanying table, obtained from the circAtlas 3.0 database.

**Figure 3 cells-14-00312-f003:**
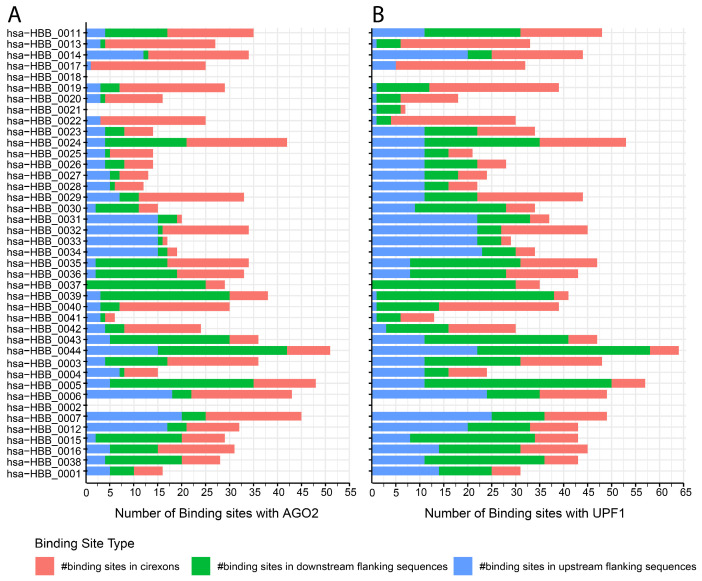
Binding site distribution for AGO2 and UPF1 in circRNAs derived from the *HBB* gene. A stacked bar chart illustrating circRNA and RBP binding site numbers: (**A**) AGO2. (**B**) UPF1. In circRNAs, upstream and downstream refer to positions relative to the back-splice junction, as they lack traditional 5′ and 3′ ends. These binding sites in flanking sequences play a crucial role in circRNA biogenesis, RBP interactions, and stability. Data obtained from circAtlas 3.0 database. (# indicate ‘the numbers’).

**Figure 4 cells-14-00312-f004:**
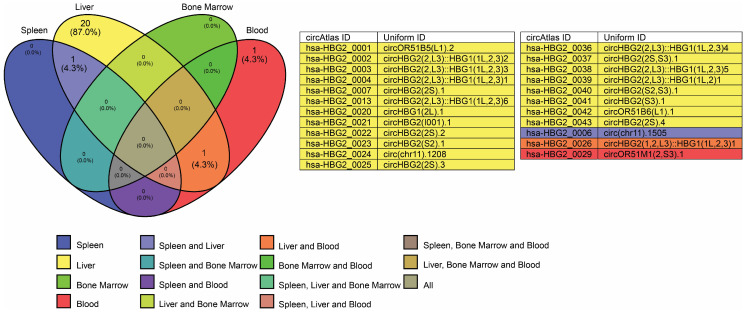
Venn diagram illustrating circRNA expression from the *HBG2* gene in blood, bone marrow, liver, and spleen. A total of 23 circRNAs are expressed across these tissues, with 1 (4.3%) in blood, 20 (87%) in liver, 1 (4.3%) in both blood and liver, and 1 (4.3%) shared between liver and spleen. The corresponding circRNA IDs and uniform IDs are obtained from the circAtlas 3.0 database.

**Figure 5 cells-14-00312-f005:**
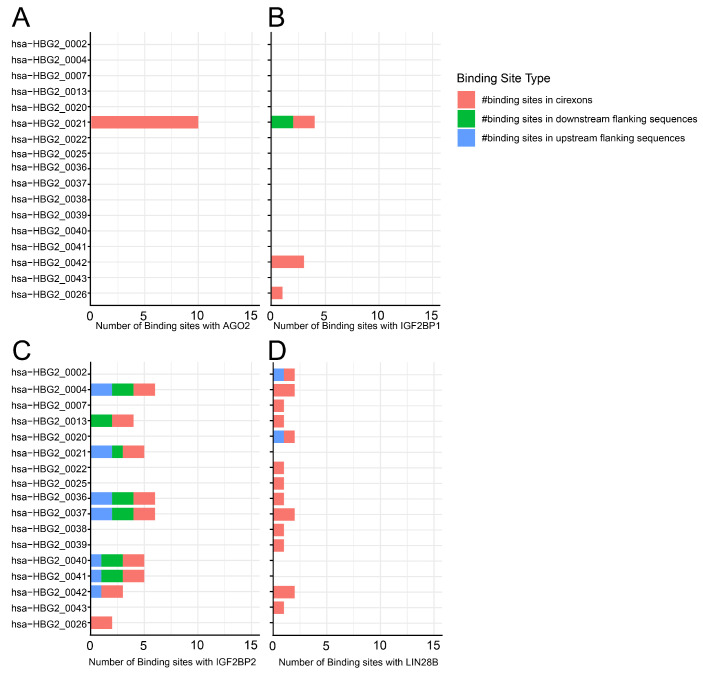
Binding site distribution for RBP in circRNAs derived from the *HBG2* gene. A stacked bar chart illustrating circRNA and RBP binding site numbers: (**A**) AGO2, (**B**) IGF2BP1, (**C**) IGF2BP2, and (**D**) LIN28B. Data obtained from circAtlas 3.0 database.(# indicate ‘the numbers’).

**Figure 6 cells-14-00312-f006:**
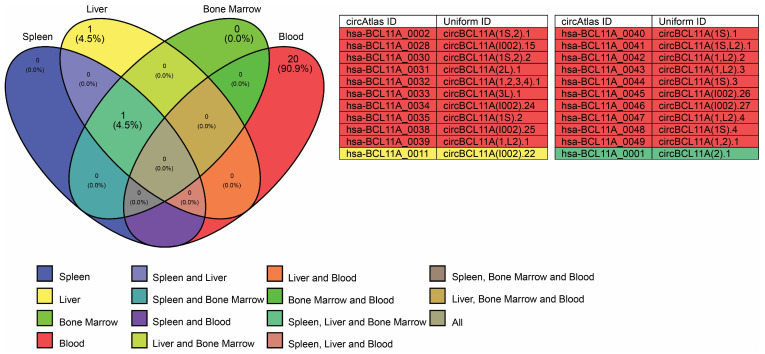
Venn diagram illustrating circRNA expression from the *BCL11A* gene in blood, bone marrow, liver, and spleen. A total of 22 circRNAs are expressed across these tissues, with 20 (90.9%) in blood, 1 (4.5%) in liver, and 1 (4.5%) shared among spleen, liver, and bone marrow. The corresponding circRNA IDs and uniform IDs are obtained from the circAtlas 3.0 database.

**Figure 7 cells-14-00312-f007:**
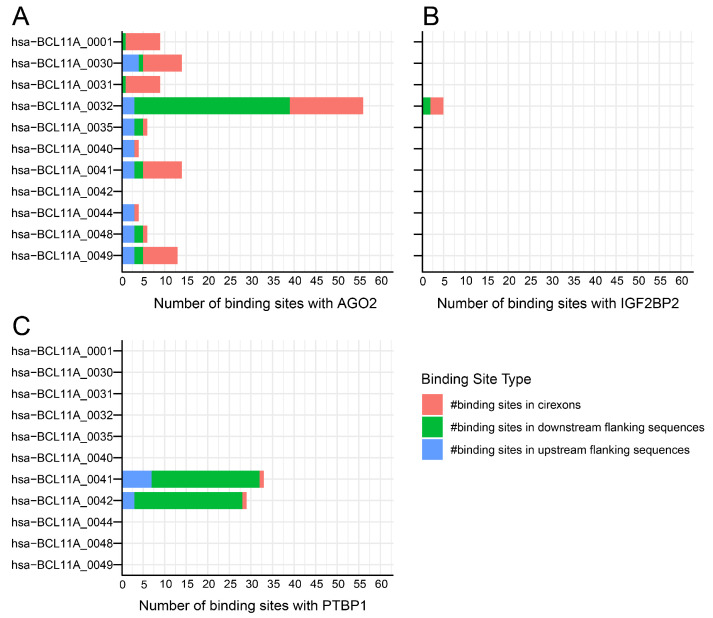
Binding site distribution for RBP in circRNAs derived from the *BCL11A* gene. A stacked bar chart illustrating circRNA and RPB binding site numbers: (**A**) AGO2, (**B**) IGF2BP2, and (**C**) PTBP1. PTBP1, an RNA-binding protein, regulates mRNA post-transcriptionally and facilitates IRES-mediated translation. Data obtained from circAtlas 3.0 database. (# indicate the numbers).

**Figure 8 cells-14-00312-f008:**
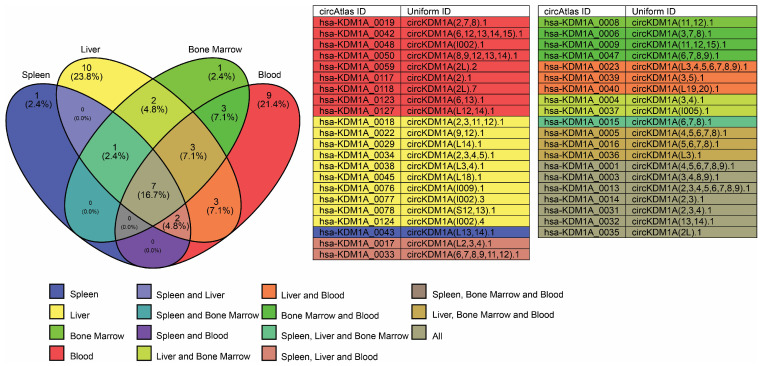
Venn diagram illustrating circRNA expression from the *LSD1* gene in blood, bone marrow, liver, and spleen. A total of 42 circRNAs are expressed, with 9 (21.4%) in blood; 1 (2.4%) in bone marrow; 10 (23.8%) in liver; 1 (2.4%) in spleen; 3 (7.1%) shared between blood and bone marrow; 2 (4.8%) between bone marrow and liver; 3 (7.1%) between blood, bone marrow, and liver; 1 (2.4%) between spleen, liver, and bone marrow; 3 (7.1%) between blood and liver; 2 (4.8%) between spleen, liver, and blood; 7 (16.7%) in all tissues. The corresponding circRNA IDs and uniform IDs are obtained from the circAtlas 3.0 database.

**Figure 9 cells-14-00312-f009:**
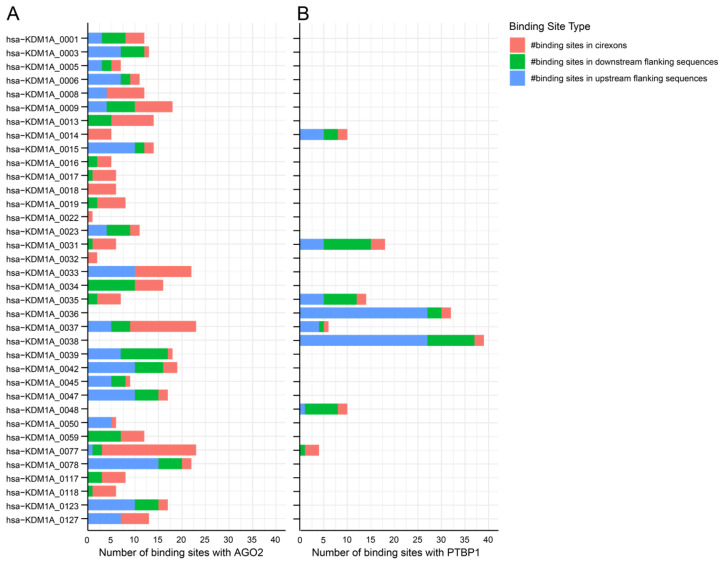
Binding site distribution for AGO2 and PTBP1 in circRNAs derived from the *LSD1* gene. A stacked bar plot illustrating circRNA and RBP binding site numbers: (**A**) AGO2 and (**B**) PTBP1. Data obtained from circAtlas 3.0 database. (# indicate ‘the numbers’).

**Figure 10 cells-14-00312-f010:**
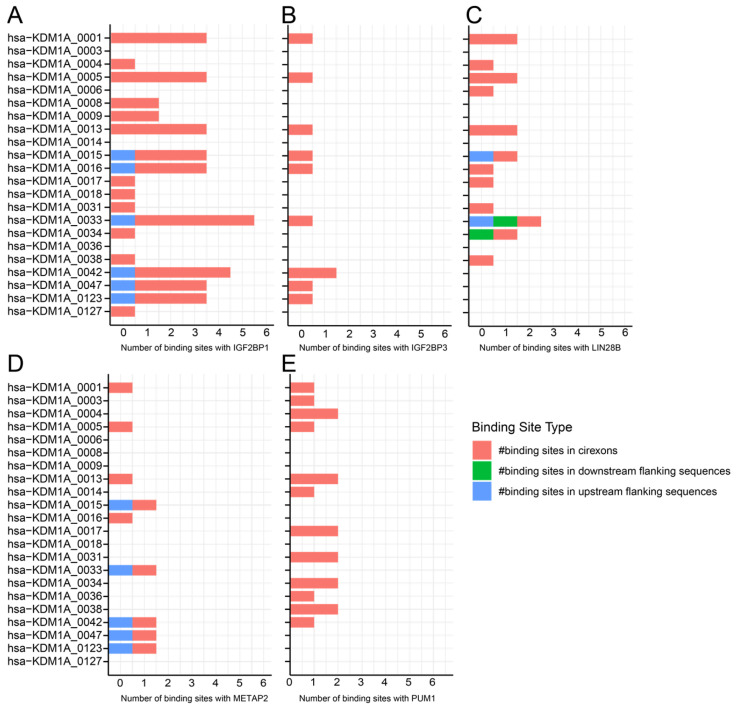
Binding site distribution for RBP in circRNAs derived from the *LSD1* gene. A stacked bar plot illustrates circRNA and RBP binding site numbers: (**A**) IGF2BP1, (**B**) IGF2BP3, (**C**) LIN28B, (**D**) METAP2, and (**E**) PUM1. Data obtained from circAtlas 3.0 database. (# indicate ‘the numbers’).

**Figure 11 cells-14-00312-f011:**
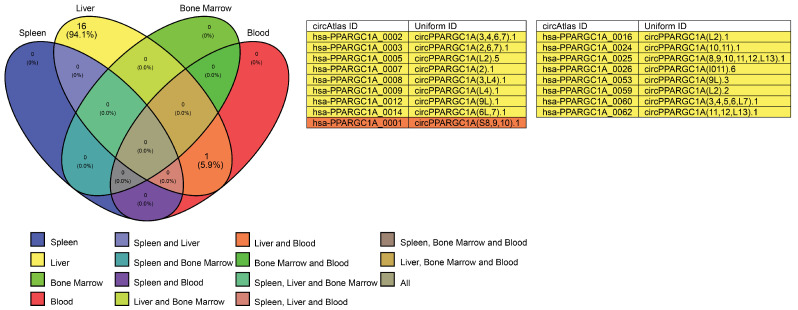
Venn diagram illustrating circRNA expression from the *PGC-1*α gene in blood, bone marrow, liver, and spleen. A total of 16 circRNAs are expressed in the liver, with one also detected in both blood and liver. The corresponding circRNA IDs and uniform IDs are obtained from the circAtlas 3.0 database.

**Figure 12 cells-14-00312-f012:**
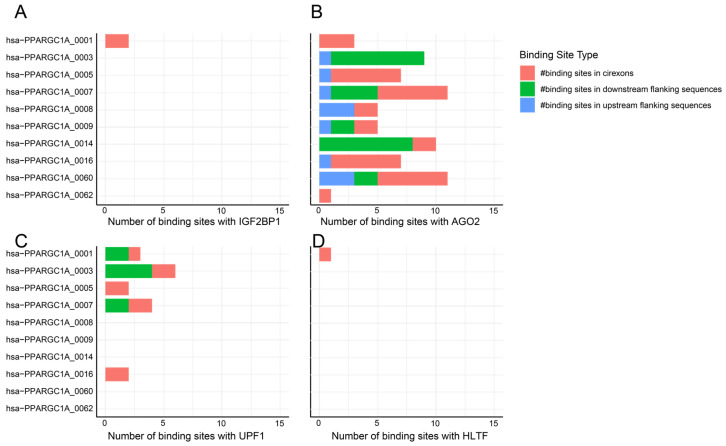
Binding site distribution for RBP in circRNAs derived from the *PGC-1*α gene. A stacked bar plot illustrating circRNA and RBP binding site numbers: (**A**) IGF2BP1, (**B**) AGO2, (**C**) UPF1, and (**D**) HLTF. Data obtained from circAtlas 3.0 database. (# indicate ‘the numbers’).

## Data Availability

The source of data for circRNA presented in the study is openly available in circAtlas, a publicly accessible circRNA database, available at https://ngdc.cncb.ac.cn/circatlas/.

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
