# Peer review of "Exploratory Review and In Silico Insights into circRNA and RNA-Binding Protein Roles in γ-Globin to β-Globin Switching"

_cells, 2025, doi:10.3390/cells14040312_

Round 1
Reviewer 1 Report
Comments and Suggestions for Authors
This review paper provides a thorough and well-written summary of the current understanding of circRNAs in gamma-to-beta globin switching. The paper is a valuable resource for researchers, clinicians, and anyone interested in the emerging field of RNA-based diagnostics. I have following minor comments:
1) The review could benefit from more extensive discussions on globin switching in a separate section. Including a figure that illustrates the mechanisms of globin switching, along with an overview of the key regulators involved, would significantly enhance the clarity and depth of the discussion.
2) I would suggest a detailed discussion on the limitations and challenges facing the field.
3) This review could be improved by a discussion on the translational aspects, including potential clinical trial data or ongoing research that investigates the direct application of circRNAs as diagnostic tools or therapeutic agents.
Author Response
I sincerely appreciate the reviewer’s thoughtful and constructive feedback. Your insightful comments have helped enhance the clarity and accuracy of the manuscript. I value your expertise and the time you have taken to provide these suggestions, which have strengthened the overall quality of this work.
Comment 1
The review could benefit from more extensive discussions on globin switching in a separate section. Including a figure that illustrates the mechanisms of globin switching, along with an overview of the key regulators involved, would significantly enhance the clarity and depth of the discussion.
Response 1
Thank you for your valuable suggestion. I agree that a dedicated section on globin switching, accompanied by a figure illustrating its mechanisms and key regulators, would enhance the clarity and depth of the discussion. I incorporate this as section 2 ( Key transcription factors regulating γ- to β-globin switching) after the introduction.
Comment 2
I would suggest a detailed discussion on the limitations and challenges facing the field.
Response 2
I am thankful for the valuable suggestion by the respected reviewer. I agree, and a new section was added titled (10 Limitations and challenges facing circRNA research)
comment 3
This review could be improved by a discussion on the translational aspects, including potential clinical trial data or ongoing research that investigates the direct application of circRNAs as diagnostic tools or therapeutic agents.
Response 3
I agree with the respected reviewer. A new section was added titled (11-Potential application of circRNA in clinical trial and diagnostic)
Reviewer 2 Report
Comments and Suggestions for Authors
This is a clearly written review article that serves as an "annotated catalog" of the potential circRNAs involved in globin gene switching. It is reasonably comprehensive, the illustrations are useful and informative, and in general there are only some relatively minor issues that need to be addressed.
1. In the initial paragraph of Section 2, the author should insert a brief discussion and explanation of the direct back splicing and the lariat intermediate models for circRNA formation.
There are also a few small but important areas where terminology should be corrected.
1. On line 55, it would be more accurate to say "treating" rather than "curing".
2. On line 100, the term "This work…" is confusing. I believe the authors are referring to enhanced understanding of the contribution of circRNA to hemoglobinopathy or thalassemia therapy, and it should be reworded to be more accurate.
Author Response
I sincerely appreciate the reviewer’s thoughtful and constructive feedback. Your insightful comments have helped enhance the clarity and accuracy of the manuscript. I value your expertise and the time you have taken to provide these suggestions, which have strengthened the overall quality of this work.
Comment 1
In the initial paragraph of Section 2, the author should insert a brief discussion and explanation of the direct back splicing and the lariat intermediate models for circRNA formation.
Response 1
I sincerely appreciate the reviewer’s valuable suggestion. To address this, a brief discussion and explanation of the direct back-splicing and lariat intermediate models for circRNA formation have been incorporated into the initial paragraph of Section 2. The revised text now reads: "In the lariat intermediate model, linear RNA is initially produced through canonical splicing, leaving behind a lariat structure that contains introns. In some cases, this lariat may also include skipped exons along with the introns. This lariat then undergoes further processing to form circRNA. In contrast, the direct back-splicing model bypasses this intermediate step, producing circRNA directly."
Comment 2
On line 55, it would be more accurate to say "treating" rather than "curing".
Response 2
I am thankful for the reviewer’s suggestion. The wording for that sentence has been changed and now read in line 53 "This groundbreaking therapy is FDA-approved to treat both sickle cell disease and transfusion-dependent β-thalassemia, marking a significant milestone in hemoglobinopathy management "
Comment 3
On line 100, the term "This work…" is confusing. I believe the authors are referring to enhanced understanding of the contribution of circRNA to hemoglobinopathy or thalassemia therapy, and it should be reworded to be more accurate.
Response 3
I sincerely appreciate the reviewer’s insightful feedback. To improve clarity and accuracy, the wording on line 98 has been revised as follows: "This review enhances our understanding of the role of circRNA in hemoglobinopathies and its potential implications for gene therapy strategies aimed at reactivating HbF production."